# TAO—The Taishan Antineutrino Observatory

**Hans Theodor Josef Steiger** [1,2]  **on behalf of the JUNO Collaboration**

1    Cluster of Excellence PRISMA+, Staudingerweg 9, 55128 Mainz, Germany; hsteiger@uni-mainz.de
2    Institute of Physics, Johannes Gutenberg University Mainz, Staudingerweg 7, 55128 Mainz, Germany

**Abstract:** The Taishan Antineutrino Observatory (TAO or JUNO-TAO) is a satellite detector for the Jiangmen Underground Neutrino Observatory (JUNO). JUNO aims at simultaneously probing the two main frequencies of three-flavor neutrino oscillations, as well as their interference related to the mass ordering, at a distance of ~53 km from two powerful nuclear reactor complexes in China. Located near the Taishan-1 reactor, TAO independently measures the antineutrino energy spectrum of the reactor with unprecedented energy resolution. The TAO experiment will realize a neutrino detection rate of about 2000 per day. In order to achieve its goals, TAO is relying on cutting-edge technology, both in photosensor and liquid scintillator (LS) development which is expected to have an impact on future neutrino and Dark Matter detectors. In this paper, the design of the TAO detector with a special focus on calorimetry is discussed. In addition, an overview of the progress currently being made in the R&D for a photosensor and LS technology in the frame of the TAO project will be presented.

**Keywords:** JUNO; JUNO-TAO; neutrino detectors; reactor neutrinos

## 1. Science Case and Motivation

JUNO aims at simultaneously probing the two main frequencies of three-flavor neutrino oscillations, as well as their interference related to the mass ordering, at a distance of ~53 km from two powerful nuclear reactor complexes in China [1]. The present information on the reactor spectra is not meeting the requirements of an experiment like JUNO, with a design resolution of 3% at 1 MeV. Unknown fine structures in the reactor spectrum might cause severe uncertainties, which could even make the interpretation of JUNO's reactor neutrino data impossible. TAO is aiming for a measurement of the reactor neutrino spectrum at very low distances (<30 m) to the 4.6 GW$_{\text{th}}$ strong core with a groundbreaking resolution better than 2% at 1 MeV [2]. Furthermore, TAO will make a major contribution to the investigation of the so-called reactor anomaly [3]. Present calculations of the reactor neutrino spectrum indicate a deficit of approx. 3% in the measured reactor fluxes. Currently, these anomalies can be interpreted as indications of the existence of right-handed sterile neutrinos. Nevertheless, it should be mentioned that recent refinements of the reactor flux models have been able to reduce this tension [4]. Beyond that, the reactor neutrino spectra recorded by Double Chooz [5], Reno [6] and Daya Bay [7] show an excess in the neutrino flux from 5 MeV to 6 MeV of unknown origin. This can be considered one of the most puzzling questions in the physics of reactor neutrinos today. Beyond these studies, TAO will search for signatures of sterile neutrinos in the mass range of 1 eV, which have just regained importance in light of the recently reconfirmed gallium anomaly by the BEST Experiment [8]. An additional goal of the TAO experiment is the verification of the detector technology for reactor monitoring and safeguard applications for the future effective fight against the proliferation of nuclear weapons material.

## 2. The JUNO—TAO Detector Design

The TAO experiment (see Figure 1) will realize a neutrino detection rate via the inverse beta decay (IBD) of about 2000 per day, which is approximately 30 times the rate in the

JUNO main detector [9]. In order to achieve its goals, TAO is relying on cutting-edge technology, both in photosensor and liquid scintillator (LS) development which is expected to have an impact on future neutrino and Dark Matter detectors. The experiment will realize an optical coverage of the 2.8 tons of Gd-loaded LS close to 95% with novel silicon photomultipliers (SiPMs), with a photon detection efficiency (PDE) above 50%. To efficiently reduce the dark count of these light sensors, the entire detector will be cooled down to $-50\,^\circ$C. The combination of SiPMs with cold LS will lead to an increase in the photoelectron yield by a factor of 4.5 compared to the JUNO central detector [9].

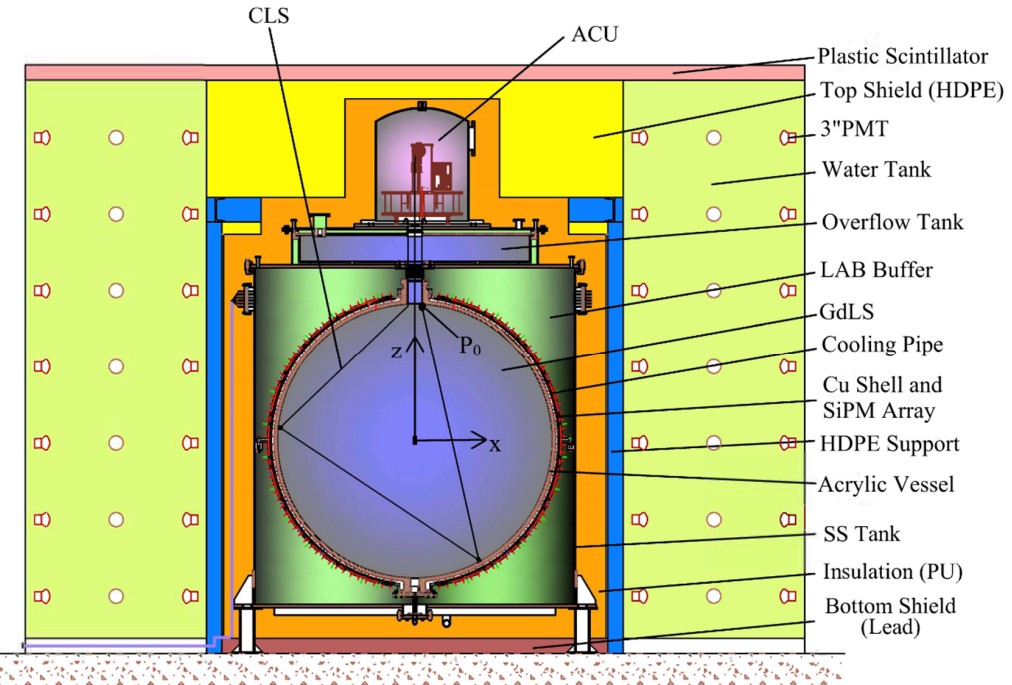

**Figure 1.** Conceptual design of the TAO detector, which consists of a central detector with the LS neutrino target and a buffer liquid, a calibration system, an outer shielding, and a veto system. The ambient and cosmogenic background will be reduced by the veto system based on plastic scintillator detectors. Layers of high-density polyethylene (HDPE) and polyurethane (PU) will act as passive shielding. A water tank equipped with PMTs acts as a surrounding water Cherenkov detector. The buffer vessel containing LAB and the inner detector (1.8 m acrylic vessel) filled with 2.8 tons of novel LAB based Gd-loaded LS as well as a 10 m$^2$ array of ~4000 SiPMs [8,9] mounted on a copper shell will be cooled down to $-50\,^\circ$C. The complete inner detector is housed in a stainless steel (SS) tank. Insulation is realized by means of PU layers. The calibration system consists of the Automated Calibration Unit (ACU) and a Cable Loop System (CLS). A segment of the CLS cable is located in the GdLS where P0 marks its starting point. A coordinate system (see x and z) is defined for the detector with its origin in the center of the target vessel. Figure taken from [10].

## 3. Cold Gd-loaded Liquid Scintillator

The TAO detector will use a Gadolinium-loaded Liquid Scintillator (GdLS) as the target material for the electron antineutrinos undergoing the IBD reaction

$$\overline{v}_e + p \rightarrow e^+ + n$$

on a proton of the scintillator. While the prompt positron signal is exploited mainly for event energy and vertex reconstruction, the neutron capture on Gd will be used as a clean well distinguishable delayed signal to reduce the accidental background [9]. While the neutron capture on hydrogen in the LS has a (n, $\gamma$) cross-section of ~0.332 barns and the energy of the emitted gamma is 2.2 MeV the advantage of adding natural Gd with

~49,000 barns and a gamma cascade of a total energy ~8 MeV is obvious. Furthermore, the neutron-capture time in the LS is significantly shortened to ~28 µs for loading with 0.1% Gd (by mass), as compared to ~200 µs in the unloaded scintillator [11].

As a consequence of the detector design and the necessity to lower the dark noise of the SiPMs, the GdLS and the buffer liquid will be cooled down to −50 °C or lower. Linear Alkylbenzene (LAB) similar to the one used in Daya Bay [11] and JUNO [12] will be the solvent of the liquid scintillator. Especially the high flash point > 130 °C and low volatility make LAB very suitable for using it in close proximity to a nuclear reactor. Commercially available LAB is mainly a mixture of molecules with 9 to 14 carbon atoms in the linear chain. It has a freezing point below −60 °C. Nonetheless, the LAB's water content may precipitate at low temperatures greatly reducing the LS's transparency. Therefore, extensive drying of the solvent is required. Furthermore, the solubility of fluors and wavelength shifters is greatly reduced at low temperatures. By adding Dipropylenglykol-n-butylether (DPnB) in sub-percent quantities as a freezing inhibitor and antioxidant, the latter problem is cured. Currently, a fluor (2,5-Diphenyloxazole, PPO) concentration of 3 g/L and 2 mg/L of the secondary wavelength shifter BisMSB (1,4-Bis (2-methylstyryl) benzene) are considered as the baseline option for TAO. New R&D for JUNO using one Daya Bay detector [12,13] shows that 1 to 3 mg/L of BisMSB will result in the highest light output for the JUNO main detector. Slightly dependent on the solvent's purity, for most detectors of various sizes, 3 mg/L BisMSB is sufficient to reach the optimal photoelectron yield. For the TAO scintillation cocktail, the emission spectrum is dominated by BisMSB (shown in Figure 2) matches well the spectral detection efficiency maximum (>50%) at ~430 nm. For TAO the solvent is loaded via the procedure developed by Daya Bay [11] using the Gd-complex with the ligand 3,5,5-trimethylhexanoic acid (TMHA). A final mass concentration of 0.1% Gd in the liquid scintillator is foreseen.

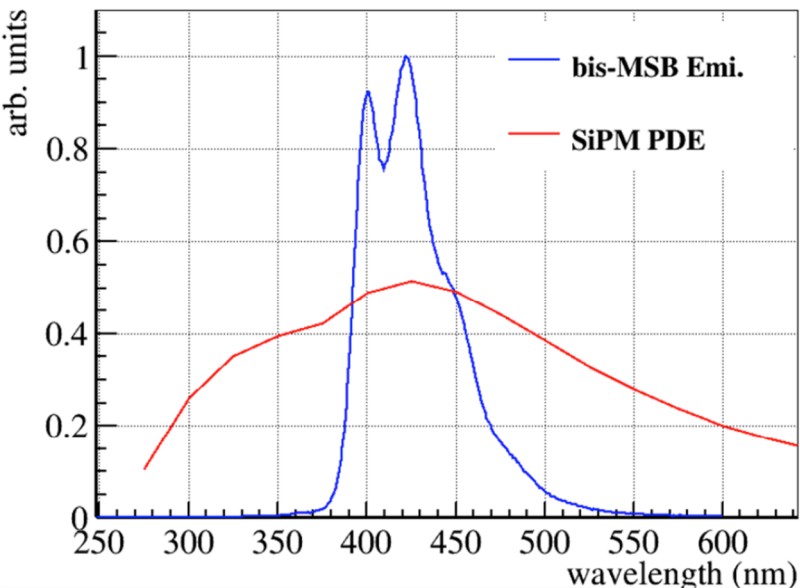

**Figure 2.** The effective emission spectrum of BisMSB (blue) dissolved in cyclohexane under excitation with UV light in a 10 mm × 10 mm × 40 mm quartz glass cuvette. The red graph represents a typical spectral response spectrum of a TAO-like SiPM at −50 °C matching well the BisMSB emission.

## 4. Calibration System

To achieve its goals, the absolute energy scale, nonlinearities, position dependencies, and detector resolution have to be understood precisely. Therefore, meticulous calibration is crucial. The calibration system contains the Automated Calibration Unit (ACU) which is reused and modified from the Daya Bay experiment [14] and a Cable Loop System (CLS), as shown in Figure 1. While the ACU (see Figure 3) can be used to calibrate TAO's energy

response precisely on the Z-axis, the CLS will allow off-axis calibrations. Moreover, the ACU contains a system based on a pulsed UV LED light source enabling timing calibration.

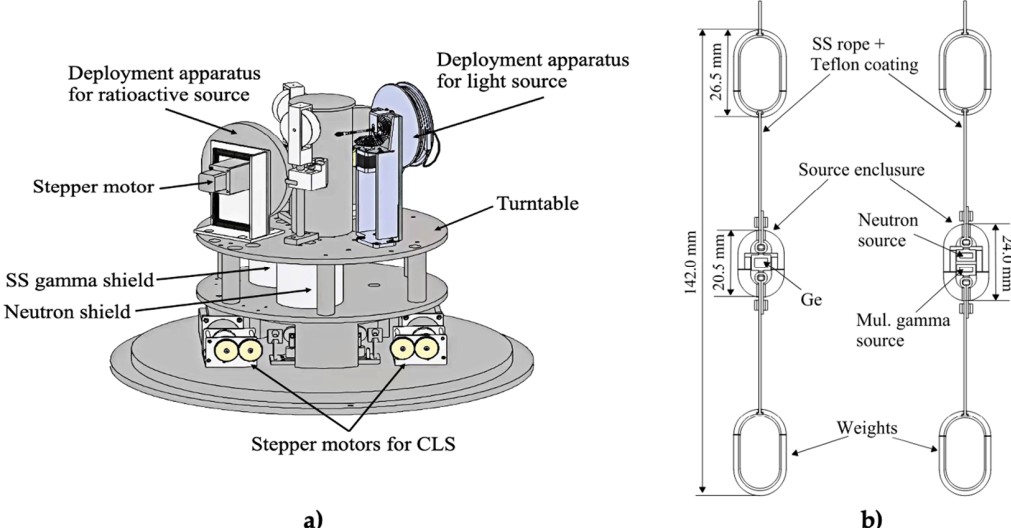

**Figure 3.** (**a**) Automated calibration system as used before in the Daya Bay Experiment [14]. The steel dome housing the ACU on top of the detector is not shown here. (**b**) Schematic drawing of the source holding structure. The $^{68}$Ge source (left) and a combined radioactive source (right). The combined source contains multiple $\gamma$ emitting isotopes ($^{137}$Cs, $^{54}$Mn, $^{40}$K, $^{60}$Co) and one neutron source ($^{241}$Am–$^{13}$C). Figures taken from [10] and modified.

### 4.1. ACU

The entire ACU (see drawing in Figure 3a) is placed below a gas and liquid tight stainless steel dome, which provides a fully enclosed system. The ACU is equipped with a turntable with three possible positions for the source deployment wheels. By that, different calibration sources (see Figure 3b) can be deployed into the detector without opening the sealed steel dome. For the TAO two of this deployment, systems will be used for radioactive sources mounted in dedicated capsules and held by PTFE-coated stainless-steel wires. The third wheel is foreseen to be used for the UV light injection system. All three deployment systems feature a wire load monitoring device, which is able to stop the motors of the ACU automatically before a source holding cable breaks in case of sticking somewhere in the detector [10,14].

### 4.2. Ultraviolet LED Calibration System

The UV light source of the ACU is equipped with a diffuser improving the isotropy of the emission. The wavelength of the UV light is (265 ± 5) nm by default but is changeable to 420 nm or any other values if necessary. The light source can be used to monitor the performance and fundamental properties of the TAO detector. This includes monitoring the state of each channel and calibrating its timing, SiPM gain, and quantum efficiency. Moreover, the UV source can be used to test the data acquisition and offline analysis pipeline. A detailed study of the central detector's pileup is feasible and foreseen.

Such a wide use of the UV LED calibration subsystem requires a highly specific design of all components. The driver of the nanosecond flasher (LED driver) applies Kapustinsky's concept [15]. Two consecutive signals are generated by two LED channels in the flasher. Increasing the amplitude of a single output signal by merging the first and second signals with each other is also possible. Nonetheless, both LEDs work independently and the respective light pulses can be set to different intensities. This is achieved by programming the steering board (see Figure 4) with its dedicated microcontroller. There also the repetition rate can be adjusted. The resulting output is monitored pulse to pulse with a control line. Therefore, the two signals are merged into a single time sequence and subsequently copied

by using an X-shape combiner-splitter. One of its outputs is guided to the detector while the other one is furnished into the control line. The latter signal is measured with a Photomultiplier Tube (Hamamatsu H7732P-01) which is connected to TAO's DAQ. The design outlined here is the development of the concept proposed in references [16,17] and realized with some changes in the JUNO Laser Calibration System [18]. Full documentation of the system is published in [10].

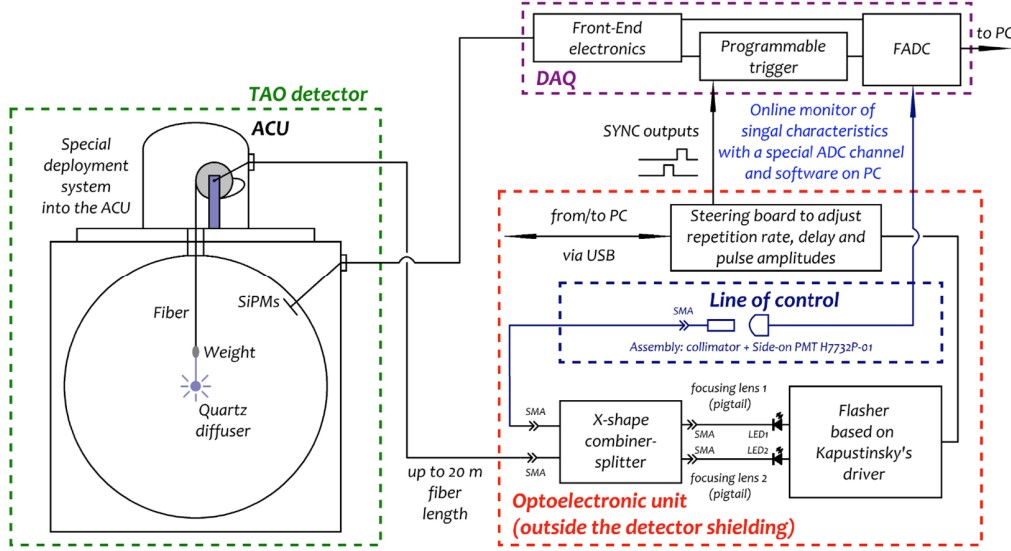

**Figure 4.** Simplified schematics of the UV LED calibration system. The dashed boxes show structural parts of the experimental setup namely the detector (green), the DAQ electronics (purple), and the optoelectronic unit (red) of the UV LED subsystem. The latter generates light pulses which are transferred to the detector target via fiber optics. Simultaneously, the DAQ is triggered from the steering board of the LED subsystem. Subsequently, the ADC digitizes also the signals from the control line. Figure taken from [10].

### 4.3. CLS

The Cable Loop System (CLS) uses one radioactive source ($^{137}$Cs), that can be deployed to an off-axis position in the target of TAO. The system development was based on experiences from a similar but drastically larger system in JUNO [19]. The radioisotope is positioned in a small area of the stainless steel cable, which is coated with PTFE along its entire length. This should prevent contamination of the Gadolinium-loaded scintillator and makes cleaning of the cable simpler. Glued anchors on the inner surface of the acrylic vessel guide the cable within the detector. Two stepper motors are used for pulling it in either direction to position the radioactive source in the target with high accuracy.

## 5. Conclusions

JUNO aims at simultaneously probing the two main frequencies of three-flavor neutrino oscillations, as well as their interference related to the mass ordering. The present information on the reactor spectra is not meeting the requirements of a high-resolution experiment like JUNO. Therefore, the TAO experiment aims for a measurement of the reactor neutrino spectrum with the unprecedented resolution of 2% at 1 MeV to identify unknown fine structures. Furthermore, TAO will make a major contribution to the investigation of the so-called reactor anomaly. Beyond that, the reactor neutrino spectra recorded by Double Chooz, Reno, and Daya Bay show an excess in the neutrino flux from 5 MeV to 6 MeV of unknown origin. By its excellent statistics and resolution, the TAO experiment can be expected to shed light on this excess and clarify if it is caused by non-standard neutrino interactions.

TAO will be built directly outside the containment of the new reactor core Taishan 1, which is one of the strongest nuclear reactors in the world. In order to realize its goals, TAO

relies on a completely new concept for liquid scintillator detectors. The experiment will realize an optical coverage of the 2.6 tons of Gd-loaded LS close to 95% with novel SiPMs with a photon detection efficiency above 50%. Cooling down the detector to $-50\,^{\circ}$C will allow a drastic reduction in the SiPM's dark noise. The combination of SiPMs with cold LS will lead to an increase in the photoelectron yield by a factor of 4.5 compared to the JUNO central detector.

The chemical development of a gadolinium-loaded and cold-resistant scintillator is of essential interest for TAO. Extensive R&D projects aiming toward the optimization and full characterization of the scintillation performance at low temperatures are ongoing. However, the use of cold scintillators is not limited to TAO: Due to the high light yield, an interesting future application might be the detection of coherent neutrino-nucleon scattering on carbon in the scintillator. While intrinsic decays of $^{14}$C define a lower energy threshold of ~150 keV, neutrinos from nuclear reactors, stopped-pion sources and supernovae would produce nuclear recoil signals above this threshold. Naturally, a cold scintillator might be interesting as well for other experiments where excellent energy resolution or sub-nanosecond timing is required.

To achieve TAO's goals, the absolute energy scale, nonlinearities, position dependencies and detector resolution have to be determined with high accuracy. Therefore, meticulous calibration is crucial. Thanks to the ACU and CLS it will be possible to deploy different radioactive sources on and off the central axis of the detector mapping its response. The degradation of the energy resolution and energy scale uncertainty will be controlled by 0.05% and 0.3%, respectively, making TAO able to measure the electron antineutrino spectrum of a nuclear reactor with unprecedented precision.

**Funding:** This work is supported by the National Natural Science Foundation of China under Grant Number 12022505, 11875282 and 11775247, the joint RSF-NSFC project under Grant Number 12061131008, the Youth Innovation Promotion Association of Chinese Academy of Sciences, and by the Program of the Ministry of Education and Science of the Russian Federation for higher education establishments with project Number FZWG-2020-0032 (2019-1569). Beyond that, this work is supported by the Cluster of Excellence PRISMA+ at the Johannes Gutenberg-University Mainz.

**Data Availability Statement:** Not applicable.

**Conflicts of Interest:** The authors declare no conflict of interest.

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
