# Peer review of "TAO—The Taishan Antineutrino Observatory"

_instruments, doi:10.3390/instruments6040050_

Round 1

Reviewer 1 Report

Very interesting paper on the Taishan Antineutrino Observatory. In my humble opinion, it can be accepted. Some typo corrections and/or style suggestions are contained in the attached file.

Author Response

Dear Reviewer 1,

Thank you very much for the positive feedback.

The corrections of the typos were done. Thank you for them. All of them were implemented.

In the Word document i kept the yellow highlighting for all the changes.

All the best,

Dr. Steiger

Reviewer 2 Report

A nice paper in my opinion, well written and I have very few comments.

L1: I find “satellite detector” a bit misleading. Complementary detector? In some sense, it seems to be the near detector.

L24: One should mention that the reactor neutrino anomaly is debatted: https://www.sciencedirect.com/science/article/pii/S0370269322001885

L76: LAB: define once for what it stands

It is an instrumentation paper but a few more words at the beginning about how one uses TAO to improve the performance of JUNO might be nice. A motivation was given but some information how this information will be used, would make it even nicer.

Author Response

Dear Reviewer 2,

thank you for your positive feedback to my paper!

Here my answers.

L1: I find “satellite detector” a bit misleading. Complementary detector? In some sense, it seems to be the near detector.

This is the official wording of the JUNO Collaboration and of the Chinese and European Funding agencies. After discussing the issue we would prefer to use "satellite detector". For sure you are right, it is the near detector of JUNO but still an independant experiment and not only a near detector.

L24: One should mention that the reactor neutrino anomaly is debatted: https://www.sciencedirect.com/science/article/pii/S0370269322001885

I will add a reference to this paper of Giunti et al. as it is a real great work. However, in the light of the recent RENO results, there is still room for the RAA.

L76: LAB: define once for what it stands

Already done as this was mentioned by Reviewer 1 as well!

I highlighted your change request in red and kept the changes done for Reviewer 1 in yellow.

Dr. Hans Steiger
